# Behavior classification: Introducing machine learning approaches for classification of sign-tracking, goal-tracking and beyond

Camille Godin¤, Frédéric Huppé-Gourgues *

École de Psychologie, Université de Moncton, Moncton, New-Brunswick, Canada

¤Current Address: School of Psychology, University of Ottawa, Ottawa, Ontario, Canada
* pfh3221@umoncton.ca

## Abstract

Classifying behaviors in research often relies on predetermined or subjective cutoff values, which can introduce inconsistencies and reduce objectivity. For example, in Pavlovian conditioning studies, rodents display diverse behaviors which can be quantified using the Pavlovian Conditioning Approach (PavCA) Index score. This score is used to categorize subjects as sign-trackers (ST), goal-trackers (GT), or intermediate (IN) groups, but the cutoff values used to distinguish these categories are often arbitrary and inconsistent. The inconsistencies stem from variability in the skewness and kurtosis of score distributions across laboratories, influenced by a range of biological and environmental factors. To address this issue, we explored two approaches to PavCA Index score classification: the k-Means classification and the derivative method. These methods determine cutoff values based on the distribution of PavCA Index scores in the sample, allowing for broader applicability to various types of behavioral data. Our results suggest that these methods, particularly the derivative method based on mean scores from the final days of conditioning, are effective tools for identifying sign-trackers and goal-trackers, especially in relatively small samples. In contrast to existing methods, our approaches provide a standardized classification framework that reflects unique distributions. Furthermore, these methods are adaptable to a researcher's specific needs, accommodating different models and sample sizes. To facilitate implementation, we provide MATLAB code for classifying subjects using both the k-Means classifier and the derivative method.

## Introduction

Classification of behaviors is an essential aspect of many areas of research. Across many areas of science, researchers often encounter distributions of observations that need to be categorized. While grouping observations into categories may seem straightforward, doing so too broadly can oversimplify their complexity. Throughout

**Data availability statement:** The data may be found on OSF. DOI: 10.17605/OSF.IO/P5DTG

**Funding:** This research was supported by the Natural Sciences and Engineering Research Council of Canada Graduate Scholarship to C.G. and by the Discovery Grant from the Natural Sciences and Engineering Research Council of Canada (NSERC RGPIN-2018-06285) to F.H.-G. The funders had no role in study design, data collection and analysis, decision to publish, or preparation of the manuscript.

**Competing interests:** The authors have declared that no competing interests exist.

the classification process, researchers might inevitably make subjective decisions that influence the outcomes. These decisions are often subtle and can occur at various stages, such as identifying and scoring behaviors, classifying subjects, or comparing groups. The criteria for classification are often predetermined, which can be practical but may not always provide the most accurate or optimal result on different samples. Alternatively, when cutoff values are selected by the researcher during the analysis, the process can lack objectivity. Together, a lack of accuracy and objectivity can compromise the reproducibility of findings in behavioral research.

As early as 1897, Pavlov presented work on the digestive system and introduced cues associated with stimuli in the environment [1]. Now known as Pavlovian conditioning, or classical conditioning, it refers to the physical pairing of a biologically potent stimulus (US), such as food, to a neutral one, such as the sound of a buzzer. Following the association of both stimuli, Pavlov noted that a dog exhibited a biological response to a formerly neutral stimulus, now considered a conditioned stimulus (CS). In consideration of the involuntary nature of classical conditioning, cues that predict a beneficial outcome produce a conditioned response and are significant in decision-making. Predictive cues act as incentive stimuli for a subset of individuals and become attractive and desirable on their own [2].

Stimuli that are considered incentives develop three specific properties due to their link with a reward: they bias attention, they themselves become sought-after, and they motivate certain behaviors [3]. The ability of incentive stimuli to motivate behaviors represents a form of adaptation, as it enhances the probability that an individual will obtain something important, for example, food or water. These stimuli may also function as temptations, encouraging behaviors that are detrimental or unhealthy for the organism [3]. For instance, when individuals navigate our modern environment, they encounter an abundance of cues signaling the availability of fat and sugar-rich foods. These cues can act as motivators for overeating, and ultimately contribute to obesity [4,5]. Other maladaptive behaviors motivated by incentive stimuli include gambling [6,7] and substance abuse [8,9]. Notably, individuals vary widely in their ability to resist temptations provoked by incentive stimuli [3].

When animals undergo Pavlovian conditioning training, a range of behaviors arises within groups. For instance, when a subject consistently moves to a food receptacle, it suggests that they see the cue only as an indicator of forthcoming food. The tendency to continuously head towards the receptacle where the US will appear is considered the goal-tracking phenotype (GT) [10]. On the other hand, some animals will continuously approach a lever (CS), even when such behaviors are not required to obtain the food (US). This is considered to belong to the sign-tracking phenotype (ST). Continued interaction with the CS indicates that it is regarded as appealing and has acquired incentive salience [11]. Both of these phenotypes accounts for approximately a third of outbred rats each, while the remaining third is regarded as an intermediate group (IN) [12]. To determine if a subject is a goal-tracker, a sign-tracker or somewhere in between, researchers typically use Pavlovian Conditioning Approach (PavCA) Index scores and cutoff values to group subjects into categories [13]. If a subject has a PavCA Index score ranging from -1 to -0.5

inclusively, it is classified as a GT phenotype. If the score is between 0.5 and 1 inclusively, it is an ST phenotype [13]. Calculating the score involves three parameters: the response bias, the probability difference, and the latency score. A subject with a score trending towards -1 signifies that it performed mostly food cup head entries without any lever presses (GT), while a subject with a score trending towards 1 indicates that it performed mostly lever presses without any head entries in the food cup (ST).

Meyer et al. [13] introduced the PavCA Index score formula as a valuable tool for quantifying individual differences in the tendency to attribute incentive salience to reward cues. This important contribution has enabled researchers in subsequent years to explore the neurobiological basis of vulnerability to substance abuse. It is noteworthy, as highlighted by Meyer et al. [13], "that […] the cut-off [values] for classifying animals is somewhat arbitrary and more or less stringent criteria may be adopted depending on the experiment" (page 5). Consequently, there is substantial variation sign- and goal-tracking in the literature regarding the cutoff values used to categorize the PavCA scores into groups. Commonly observed values include $\pm 0.3, \pm 0.33, \pm 0.4$ and $\pm 0.5$ [2,14–16], although some studies introduce variations, such as the use of $-0.53$ and $0.59$ by Lopez et al. [17], and Lesaint et al. [18], who categorize the top 30% as STs and the low 30% as GTs. Moreover, there are discrepancies regarding the duration of training days and the specific days selected to calculate a mean PavCA Index score, which impact the approach style assessment. Some researchers opt for a mean score based on the final two days of training, spanning a range of four to seven days, whereas others exclusively consider the final day for their calculations [2,16,19].

The existence of diverse cutoff values suggests that the shape of the distribution of scores fluctuates across studies, indicating potential variations in both skewness and kurtosis. Meyer et al. described GT and ST phenotypes as emerging from genetic and environmental factors, which should yield roughly equal proportions if these factors follow a normal distribution. It seems that the repetitive and structured nature of Pavlovian conditioning protocols in a laboratory setting may artificially exaggerate extreme scores, potentially transforming an originally normal distribution into a bimodal one. If the underlying neurobiological factors are skewed, this skewness propagates into the PavCA Index Score bimodal distribution, resulting in distorted group proportions. Notably, the $\pm 0.5$ cutoff from Meyer et al. was established using a population of 1878 rodents pooled from multiple vendors and colonies, creating a symmetric bimodal distribution. However, Fitzpatrick et al. [12] demonstrated that individual vendors tend to produce subjects more skewed toward either GT or ST, meaning that when researchers work with smaller datasets form a single vendor, the resulting distribution is often asymmetrically skewed rather than perfectly bimodal. As a result, researchers arbitrarily adjust cutoffs to accommodate the inherent variability across different samples.

Modern advances in statistical techniques and machine learning tools have made discrimination and classification of observations more accessible. For example, data clustering involves grouping similar observations together. A specific cluster number has to be specified beforehand to use the k-Means algorithm [20]. The k-Means employs a partitioning method to find the optimal clustering solution. It achieves classification by minimizing the sum of squared distances from input vectors to cluster centers [21]. Widely used across various areas of science, $k$-Means extracts patterns from data and helps with tasks ranging from the diagnosis of Parkinson's disease [22] to earthquake epicenter clustering [23]. In such cases, the number of clusters ($k$) needs to be predefined by the user, which directly influences the clustering results.

For a machine-learning classification method to be widely employed by researchers using the sign- and goal-tracking model, key considerations include ease of use and interpretability. Given its simplicity and intuitiveness, $k$-Means is an interesting option in the ST/GT context. However, it is essential to consider an alternative approach for classifying subjects in the ST and GT groups since unsupervised machine learning tools, including $k$-Means, have limitations. For example, the $k$-Means approach may overlook nuances or intricacies in the data as it assumes that clusters have roughly the same shape and size, and it is sensitive to outliers [24]. Alternatively, a straightforward mathematical approach could reveal the cutoff values most appropriate for the sample. Since the PavCA scores tend to distribute in a bimodal fashion in large or pooled samples [13], it is possible to extract a function representing their density distribution. The first derivative of the

density function allows the analysis of the variation in the slope parameter throughout the function, leading to the identification of maximum and minimum values of the slope. We argue that a local minimum would give a cutoff value that matches the GT distribution, while a local maximum would give a cutoff value for the ST distribution.

The aim of this study was to evaluate methods for categorizing subjects into three groups according to the distribution of their behaviors. The ST and GT model, based on the distribution of PavCA Index scores, provided a suitable framework for assessing performances of classification methods on behavioral data. We applied two accessible machine learning tools to establish objective cutoff values customized to the distribution of the sample rather than relying on typical arbitrary cutoff values for classification: 1) The k-Means classifier, which clusters PavCA Index scores into three groups using cluster edges as cutoff values, and 2) The derivative method, which determines cutoff values based on the positions of minimum and maximum slopes in a density function derived from the distribution of PavCA Index scores. These methods were selected to offer practical classification tools for researchers working with behavioral models.

To compare classification outcomes, we examined the distribution of ST, GT and IN group frequencies across classification methods, including the conventional ±0.5 cutoff values. By applying these classifiers to both a large, pooled sample and a smaller validation sample, we examined whether cutoff values remain consistent across different sample compositions. Given that the pooled sample integrates data from multiple cohorts and colonies, we hypothesized that its classification thresholds might align more closely with the conventional ±0.5 cutoff compared to the smaller validation sample.

## Method

### Animals

A total of 223 Long-Evans rats were used for this study. Rats were bred from outbred Long-Evans females purchased from Charles River (Saint-Constant, QC). Of these, 189 come from various cohorts and colonies and were trained between June 2021 and July 2022 as part of multiple experiments focused on attentional mechanisms [25,26]. The Pavlovian conditioning was conducted prior to any additional procedures. Their data was pooled to form the modeling sample, following the approach of Meyer et al. [12]. The remaining 34 rats were from the same colony (F = 13, M = 21) and were trained simultaneously in July 2022. They served as the validation sample for the model. Subjects from the validation sample were part of a feeding-schedule experiment, and the feeding-schedule results are reported in supplementary material.

Pups were separated from dams between 21 and 28 days after birth, relating to their capacity to feed independently. Given that they were still young and growing, they were given food *ad libitum* until two days prior to the beginning of Pavlovian conditioning pre-training. Once they were fully grown (around 12 weeks after birth), or at the start of Pavlovian conditioning (for standardization), food access was restricted to 60 minutes per day to ensure body weight stability during testing and avoid obesity [27]. On average, rats started PavCA conditioning between one and three months of age. During the two days preceding the initial session, each rat was given 15 neutral flavor, 45 mg food pellets (Bio-Serv; Frenchtown, New Jersey) and manipulated two minutes each by the experimenter. All rats included in the modeling sample were under a food-deprivation program that limited their access to food before doing the tasks, (Fig 1 in S2 Fig) for the effect of food deprivation. In the validation sample, 17 rats were under the food-deprivation program (F = 8, M = 9), whereas 17 rats had unlimited access to food (F = 9, M = 8). Rats selected for food deprivation were counterbalanced to avoid a cohort effect. Rats were paired in cages according to sex and housed in a climate-controlled room with an inverted light cycle. Water and food (Rodent Chow, Laboratory Rodent Diet) were freely available until the start of the training, and their corn-based litter was replaced twice a week. The Université de Moncton Animal Welfare Committee approved all procedures (Protocol No 21–02) and followed the Canadian Council on Animal Care guidelines and protocols.

## Apparatus

Six standard operant conditioning chambers were used for conditioning. Each conditioning chamber contains a floor made of stainless-steel rods, a plexiglass sheet on top and sides of the chamber, and two PLA (polylactic acid) plastic or metal front-and-back walls. The floor measures 20 cm by 30 cm and a catch tray bedding is located underneath the rods. At approximately 2.5 cm to the left or right of the food cup and 6 cm above the floor was positioned a retractable lever (Coulbourn H23-17M). To ensure lever-side counterbalancing, the rats were randomly assigned to the chambers in approximately equal numbers before the experiment and were tested in their assigned box throughout the training. Inside the lever compartment, a white LED illuminated the slot, making the lever visible when it was extended. A red house light (Stoelting, Any-Maze red cue light) on the opposite wall stayed on for the entire experiment. The pellet dispenser delivered one 45 mg neutral flavor food pellet (Bio-Serv; Frenchtown, New Jersey) into the food cup at a time. Inside the food cup, an infrared photobeam was positioned 1.5 cm above the bottom to detect the head insertion, and head entries were recorded accordingly. The boxes also had a food reward collection area, or a "feeder" (Coulbourn H14-22R-45 coupled to H14-01R). Cue presentation, lever operation, reward distribution, and data collection were computer controlled (GraphicStateRT4, Coulbourn system). A fan ensured background white noise, and each conditioning chamber was individually isolated in a noise-reducing enclosure.

## Pavlovian Conditioning Approach (PavCA) training

Pre-training sessions involved placing the rats in the operant chambers, during where the red house light remained stayed illuminated and the lever remained retracted. Before each session, we executed a short test to ensure the proper function of all operant chambers. A total of 25 food pellets were dispensed on a variable interval schedule, with an average timing of 90-seconds between deliveries. We monitored the consistency of pellets retrieval from the food cup. The pre-training sessions spanned, in average, 30 minutes. After pre-training, each rat underwent Pavlovian conditioning in the same operant chambers used for pre-training. The testing followed standard PavCA procedures [10]. Each trial of a Pavlovian conditioning session consisted of the presentation of a conditional stimulus (an illuminated lever) for eight seconds. The lever retraction was immediately followed by a food pellet being dispensed into the food cup. The food delivery was response-independent, which meant that it did not require lever activation. The next intertrial interval began immediately after the food delivery. The lever was presented at random intervals ranging from 30 to 90 seconds. Each test session consisted of approximately 25 trials, one trial being the presentation of the lever and the food pellet, resulting in an approximately 40-minute session each day. In total, each rat underwent six days of training. Each press of the lever was detected with a sensor. Food pellet consumption was recorded by an infrared beam detecting head entry in the cup. We noted the number of remaining pellets in the dispenser at the end of each session, as well as the total number of consumed pellets by subtracting the remaining number.

Animals were categorized into three groups (ST, GT and IN) using the PavCA Index score formula by Meyer et al. [13]. The formula was based on events recorded in the PavCA chambers, which were the number of lever presses, head entries into the food cup, lever press latency and head entry latency. First, we calculated the response bias, a score that reflects the relative preference for one behavior over another. A more positive score indicates a preference towards lever pressing, while a more negative score reflects a preference towards head entries. Secondly, we calculated the probability difference, which was the difference between the probability of pressing the lever and the probability of entering the food cup. Thirdly, we calculated latency score, which was the difference between the latency to approach the lever and the food cup. By averaging these three approach measures, a PavCA score was generated for each subject on every day of training. The produced scores varied from −1 to +1, with scores of −1 and +1 representing a strong bias towards GT and ST respectively, while a score equivalent to zero indicated an equal distribution of the two types of responses. For further details about the PavCA Index score formula and calculation, refer to Meyer et al. (2012).

## Statistical analysis

**Characteristics of the modeling and validation sample.** A MATLAB (MATLAB 2020b, The MathWorks, Inc., Natick, Massachusetts) routine was coded and used to extract the data from the GraphicStateRT4 Coulborn files. The recorded activity was reduced into the five variables used to calculate the PavCA Index score, namely lever contacts, food cup entries, latency score, response bias and probability difference. Parametric and nonparametric analyses were conducted using Jamovi for Windows software (version 2.0.0.0). The *95%* confidence intervals were adjusted for differences and correlations with the superb framework (version 0.95.7) [28].

The covariance structure for the repeated-measure data in the validation sample was explored and modeled for each group of the independent variable (food restriction and sexes). The significance threshold was set to $p < 0.05$. The ANOVA was conducted both on the modeling data ($n = 189$) and the validation data ($n = 34$). We executed a mixed design analysis of variance (ANOVA 2x2x6) of two fixed effect factors (sexes: males and females, food: restricted and non-restricted) and six repeated measures of PavCA scores. Since conducting covariance analyses without meeting normality assumptions can be underpowered [29], nonparametric tests were performed on the sexes and food availability as well (two Kruskal-Wallis one-way ANOVA tests).

PavCA Index scores of the 189 subjects were extracted using the modeling sample. An analysis of variance (ANOVA 6x1) of the six repeated measures was conducted on the PavCA Index scores to assess changes of scores across days. Mean scores of the last two days of training were calculated to assess each rat's tendency for ST or GT.

## Classification

**k-Means algorithm.** All classification analyses were executed using MATLAB's Statistics and Machine Learning Toolbox (code provided in Annex D). A *k*-Means cluster algorithm was trained on the PavCA Index scores of the modeling sample. Based on the three categories of classification normally used (ST, GT, IN), the *k*-Means cluster analysis was performed with $k = 3$ clusters. The *k*-Means was performed on individual and mean scores of Days 4–5 and 5–6. We also executed the same *k*-Means analysis on the combined three dimensions used to calculate the PavCA scores (latency score, lever presses, food cup entries). We present the results of the *k*-Means analysis, which includes cutoff values, centroids and dispersion, based on the PavCA Index scores from each day of conditioning.

**Derivative method.** We used an alternative method, which we called the derivative method, to determine appropriate cutoff values for the distribution. Using the *ksdensity* function from MATLAB, we obtained an estimate of the density probability of the scores. We used the default bandwidth, determined by Silverman's rule of thumb, as it provides a data-driven estimate without prior distribution assumptions. Furthermore, we used *ksdensity*'s default unbounded domain to avoid condensing and distorting estimates. The resulting density function of the PavCA score distribution for each day of conditioning can be derived to obtain the first derivative of that function. The peaks (and their absolute values) of the first derivative function are equivalent to the local maximums and minimums of the slope of the bimodal PavCA score distribution. We hypothesized that the second local maximum slope of the density function served as a suitable approximation for identifying the cutoff value that encompassed the majority of the ST group. Additionally, the first local minimum indicated the optimal position for classifying the GT group. We calculated cutoff values for individual and mean scores of Days 4–5 and 5–6.

**Comparing methods.** All three methods of classification (the typical ±0.5, the *k*-Means, and the derivative) were applied on both the modeling and validation samples for the individual and mean-scores of days 4–6 of conditioning. The McNemar tests on IBM SPSS Statistics (version 28.0.0.0) were used to compare the frequencies of observations in the ST, GT and IN groups. Given that the test was reiterated on the same sample for each method, the test's significance was predefined at $p < 0.01$. The frequencies of observations were compared for three conditions: 1) using ±0.5 cutoff values, 2) using *k*-Means cutoff values, and 3) using the derivative cutoff values.

## Results

### Characteristics of the validation sample

We conducted a mixed-design analysis of variance (ANOVA 2x2x6) on the PavCA scores to evaluate if there were PavCA Index scores differences based on the subjects' sex and training day **Fig 1**. Sphericity assumptions were not met ($W = 0.118$, $p < 0.01$). The results revealed that only one factor significantly influenced the scores ($p = 0.007$), which was the training day. A nonparametric one-way ANOVA tests (Kruskal-Wallis) was also carried out to assess the effects of sexes on PavCA Index scores. The Kruskal-Wallis test revealed that there were no significant differences between PavCA Index scores of males and females ($\chi^2 = 0.00863$, $p = 0.926$).

### Characteristics of the modeling sample

We performed an analysis of variance (ANOVA 1x6) on the PavCA Index scores of subjects on each day of conditioning. The analysis showed that the day had a significant impact on the PavCA Index scores ($F = 79.7$, $p < 0.001$, $\eta^2_p = 0.32$). Through Tukey post-hoc analyses, we observed significant differences between the PavCA Index scores of Day 1 and all other days ($p < 0.001$), and Day 2 and all other days of conditioning ($p < 0.001$). Additionally, we observed a significant difference between Day 3 and Day 6 ($p = 0.003$). **Fig 2** depicts the distribution of PavCA Index scores per days of conditioning in the modeling sample.

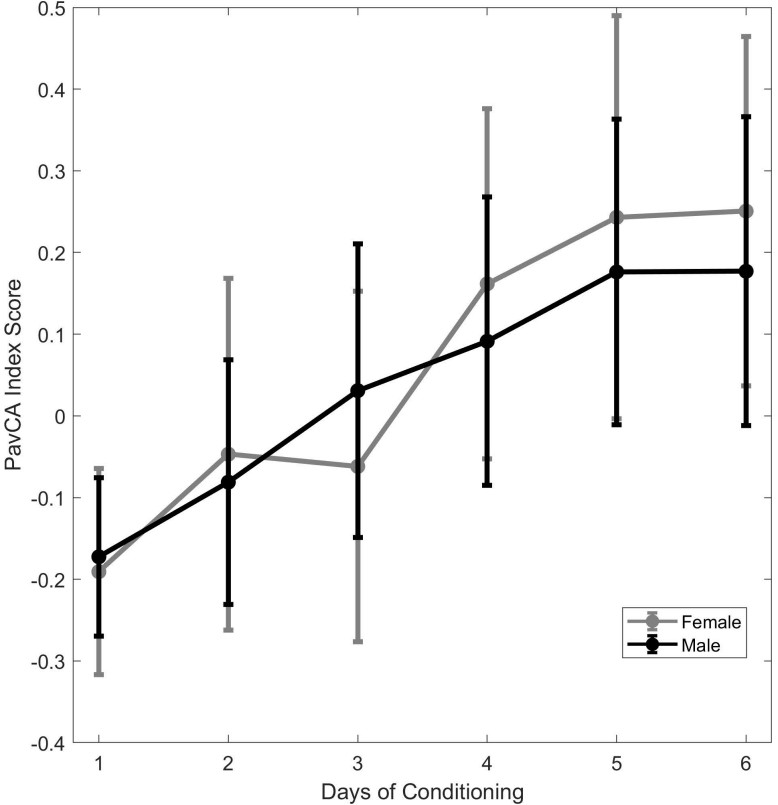

**Fig 1. PavCA Index scores per days of conditioning for males and females.** The circles illustrate the mean Pavlovian Conditioning Approach (PavCA) Index scores obtained on each day of PavCA. The error bars depict difference- and correlation-adjusted 95% Confidence Intervals (CI), computed according to the method presented by Cousineau et al. [28].

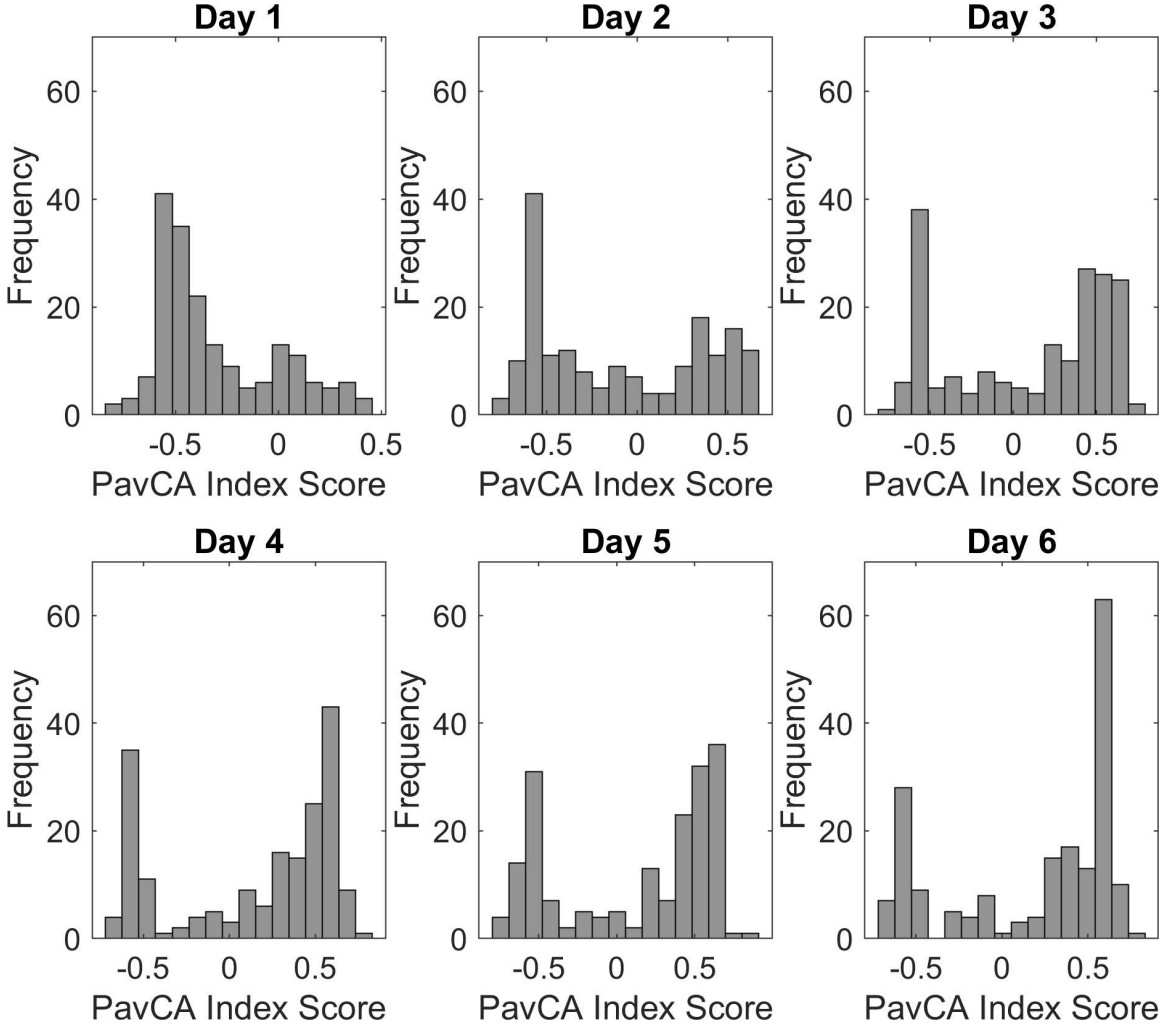

**Fig 2. Distributions of PavCA Index scores across days of conditioning in the modeling sample.** Each bar represents frequency of observations in between intervals of Pavlovian Conditioning Approach (PavCA) Index scores. Each graphic contains 10 bars, a total of 189 subject scores were classified on each day of conditioning.

## Classification with *k*-Means in 1D

The *k*-Means cluster algorithm was trained on PavCA Index scores from the modeling sample data set. To find the edges of the three PavCA Index score groups used in the sign- and goal-tracking literature, the number of clusters had to be predefined to $k = 3$. All 189 rats were trained over six days. We used the PavCA mean scores of Days 4–6 and 5–6 of the modeling sample data set to train the *k*-Means algorithm. We compared the resulting cutoff values to those extracted on Days 4, 5 and 6 separately. The cutoff values, the centroids and the point-to-centroid distances (Sum D) statistics of each day of conditioning and mean scores are shown in **Table 1**. In k-Means clustering, the centroid represents the average position of points within a cluster, obtained from the C output of the k-means function. Thus, the centroid is a center location value. Point-to-centroid distances refer to within-cluster sums of distances from individual points to their respective centroids, returned as sumd in k-means.

**Table 1. Classification of PavCA Index scores with the k-Means algorithm where k = 3.**

| Days | Cutoff Values | | Centroid | | | Sum *D* | | |
|---|---|---|---|---|---|---|---|---|
| | ST | GT | ST | GT | IN | ST | GT | IN |
| 4 | > 0.36 | <-0.17 | 0.56 | -0.54 | 0.17 | 0.63 | 0.56 | 1.15 |
| 5 | > 0.34 | <-0.22 | 0.55 | -0.55 | 0.12 | 0.80 | 0.70 | 0.79 |
| 6 | > 0.38 | <-0.17 | 0.57 | -0.53 | 0.19 | 0.57 | 0.86 | 1.21 |
| 4-5 | > 0.31 | <-0.20 | 0.53 | -0.54 | 0.10 | 0.86 | 0.56 | 0.71 |
| 5-6 | > 0.31 | <-0.21 | 0.53 | -0.54 | 0.07 | 1.01 | 0.58 | 0.70 |

PavCA mean scores of Days 4–5 and 5–6 were calculated before training *k*-Means on the mean scores.

## Classification with the derivative method

The second classification method, referred to as the derivative method, involved applying a smoothed density estimate function to the distribution of PavCA Index scores for each day of conditioning in the modeling sample. This process generated a density function that could be analyzed mathematically. By deriving the density function f($x_i$) of the distribution, we obtained *dfdx* = diff(*f*)/diff($x_i$), indicating the fluctuation of the slope parameter across the f($x_i$) curve. The local maximums and minimums on the first derivative of the density function indicated where the slope reached its maximal and minimal values, respectively. These derived values were then used to locate the associated PavCA Index scores on the density function. Since our aim was to determine cutoff values for ST and GT, the minimum value of the first parabola appropriately represented GT cutoff and the maximum value of the second parabola represented ST cutoff. Cutoff values were obtained for GT and ST for individual and mean scores of Days 4–5 and Days 5–6. The cutoff values are presented in **Table 2**, with peak locations and peak widths. The peak location is determined using the *findpeaks* function on the output function of ksdensity, where outputs peakLoc and peakWidth represent the location of the peak and the dispersion around that value, respectively. Peak widths are computed as the distance between the points to the left and right of the peak where the function intercepts a reference line. Thus, peak locations represent a center value for each group (ST and GT). **Fig 3** illustrates a density function of Day 5–6 mean scores and its derivative, with dotted lines indicating the local minimum and maximum values and their corresponding PavCA Index scores, which serve as the cutoff for groups in that distribution. For a visual depiction of cutoff values, centroids and dispersions across days of conditioning and groups, refer to supplementary materials (Fig 2 in **S2 Fig**).

## Comparing frequencies of observations - modeling sample

We tested the frequency of observations in each group (ST, GT and IN) extracted from all three methods of classification applied on the mean score of the last 2 days (days 5–6) of conditioning with McNemar's test on group proportions. The observed frequencies for each method are illustrated in **Fig 4** (A) and are as follows: k-Means: 100 ST, 50 GT, 39 IN; derivative

**Table 2. Classification of PavCA Index scores (modeling sample) with the derivative method.**

| Days | Cutoff Values | | Peak Location | | Peak Width | |
|---|---|---|---|---|---|---|
| | ST | GT | ST | GT | ST | GT |
| 4 | > 0.34 | <-0.44 | 0.54 | -0.56 | 0.52 | 0.30 |
| 5 | > 0.37 | <-0.44 | 0.55 | -0.56 | 0.45 | 0.31 |
| 6 | > 0.43 | <-0.48 | 0.57 | -0.59 | 0.41 | 0.26 |
| 4-5 | > 0.33 | <-0.44 | 0.50 | -0.56 | 0.63 | 0.30 |
| 5-6 | > 0.36 | <-0.45 | 0.53 | -0.57 | 0.51 | 0.29 |

PavCA mean scores of Days 4–5 and 5–6 were calculated before training *k*-Means on the mean scores.

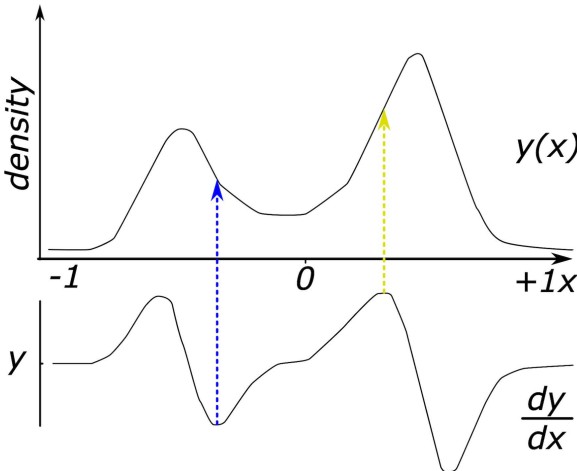

**Fig 3. Density function of mean PavCA Index scores and its derivative function.** The upper curve represents a density estimate of mean Pavlovian Conditioning Approach (PavCA) Index scores obtained from the last two days of conditioning, extracted with a smoothing function of the true distribution of scores. The lower curve depicts the first derivative of this density function. The blue line marks the local minimum of the derivative function, indicating a suitable cutoff value for goal-tracking scores. The yellow line indicates the local maximum of the derivative in an area associated to sign-tracking scores.

method = 92 ST, 40 GT, 57 IN; and ±0.5 cutoff: 65 ST, 39 GT, 85 IN. The McNemar's test results revealed that for the IN and ST groups, the *k*-Means and derivative models produced group proportions with significant differences ($p < 0.01$) from the ±0.5 cutoff values. For the GT group, the results revealed no significant difference between the derivative and the ±0.5 ($p = 0.317$). Between the *k*-Means and the ±0.5 methods for the GT group, the results revealed a significant difference ($p < 0.001$).

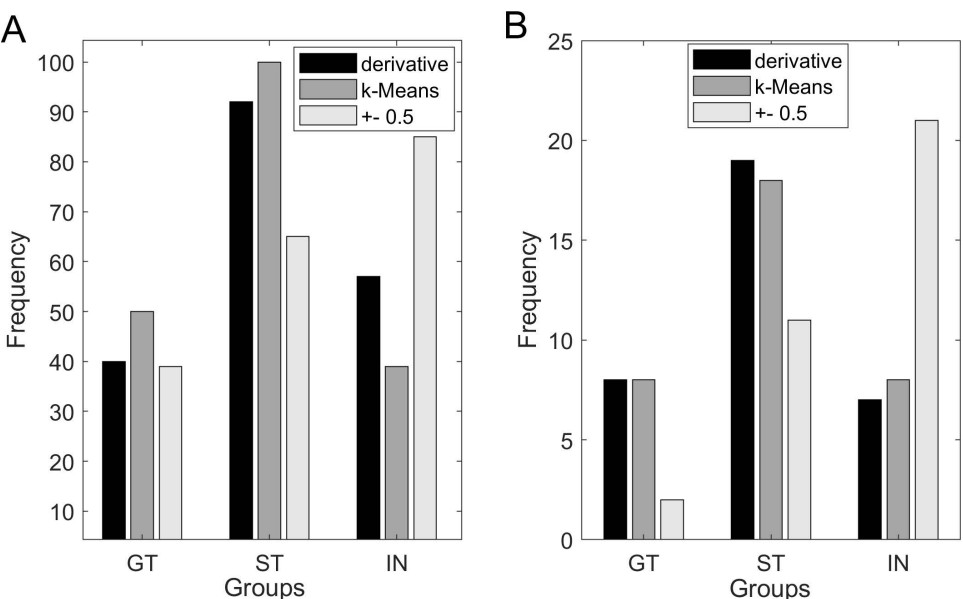

**Fig 4. Frequencies in each group according to the 3 classification methods.** ( (A) Frequencies of subjects classified in each group/phenotype with 3 classification methods in the modeling sample. (B) Validation sample. Proportions are distorted the most with the ±0.5 cutoff in the validation sample, compared to the k-Means and derivative.

When comparing the *k*-Means and the derivative methods on the mean score of the last two days of conditioning, results suggested a significant difference in the ST proportions ($p = 0.005$). For the GT, results revealed a significant difference ($p < 0.001$). For the IN group, results revealed a significant difference ($p < 0.001$). Examining Tables 1 and 2 reveals that 1) both methods of classification extract similar cutoff values, 2) the derivative method yields more conservative cutoffs, and 3) the GT group exhibits the greatest discrepancy in classification between the two methods. For a visual of the group center values, refer to asterisks in Fig 8.

The progression of subjects in each phenotype group according to the three classification methods is illustrated in **Fig 5**. Despite variation in classification, the overarching patterns of classification appear to be largely consistent across methods. This suggests that the introduction of new methods may not significantly alter the general composition of each phenotype group. Instead, using the *k*-Means or derivative methods may have a greater impact on the classification of a few individuals located in close proximity to group boundaries, though the overall group composition appears to remain unaffected.

### 3D *k*-Means classification

Given that the *k*-Means clustering algorithm can be performed on multidimensional data, the algorithm was trained on three parameters used to calculate the PavCA Index scores (instead of classifying the PavCA Index scores themselves). The three parameters, which are the latency score (difference in latencies to make food cup entries and lever responses) the number of lever presses, and the amount of food cup entries, were inserted as three variables in the function. The pooled data from the last two days of conditioning from the modelling set was used. A different distribution of scores was observed in the three respective groups. The categorization provided by the three-parameter *k*-Means clusters does not

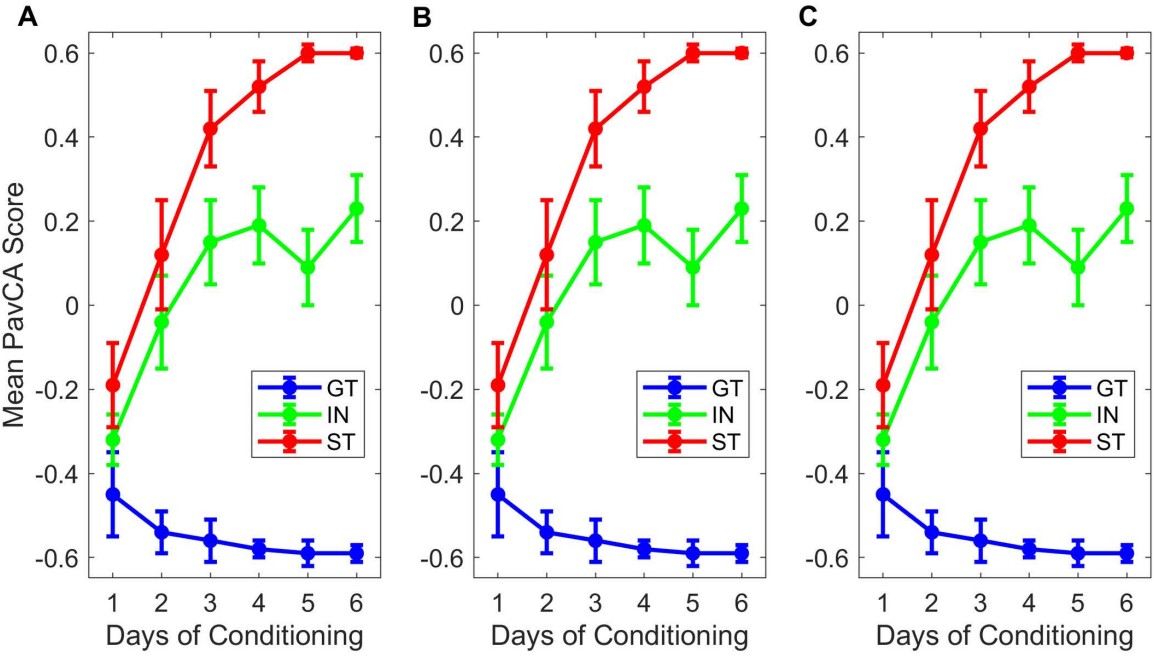

**Fig 5. Progression of scores across days of conditioning for each phenotype group according to the classification methods. (A)** K-Means results. **(B)** Derivative method. **(C)** Standard ±0.5 cutoff. Classification was performed on the mean scores of the last two days of conditioning in the modeling sample. Each curve shows phenotype groups progressing over time. The error bars illustrate difference- and correlation-adjusted 95% Confidence Intervals (CI), computed according to the method presented by Cousineau et al. [28]. Each method seems to capture similar subjects and trends, they become distinct at the edges of each phenotype.

appear to match the categorization of the unidimensional *k*-Means. Examination of **Fig 6** suggests that the groups are linearly delimited at -0.21 and 0.31 with 1D *k*-Means classification (A), whereas the same linear delimitation is not apparent for IN scores with the 3D classification (B).

As we examined the relationship between the three parameters and their associated PavCA Index scores in 3D (**Fig 7**), we observed that the data did not visually form three distinct clusters. Sign-tracker to goal-tracker scores, color-coded from red to blue, were distributed mostly through the food cup entry and lever press dimensions. However, the latency score dimension appeared to lack significant discrimination, except for a few dark-blue points associated with low latency scores and PavCA Index scores, and low-mid food cup entries.

### Classification - validation sample

Classification provided by the ± 0.5 cutoff values (Meyer et al., 2012) was compared with the *k*-Means and the derivative methods. The *k*-Means classifier and the derivative methods were applied on the validation sample to provide its cutoff values. On Days 5 and 6 the derivative method provided two cutoff values (Day 5 = GT < -0.21, ST > 0.25, Day 6 = GT < -0.23, ST > -0.38). On the same days, the *k*-Means provided the following cutoff values: Day 5 = GT < 0.0, ST > 0.40, Day 6 = GT < -0.16, ST > 0.36. To stay true to Meyer et al., the cutoff values from the mean scores of Days 5 and 6 were extracted with each classification method. The derivative method revealed the following cutoff values: GT < -0.26, ST > 0.25. The *k*-Means method revealed the following cutoff values: GT < -0.14, ST > 0.37.

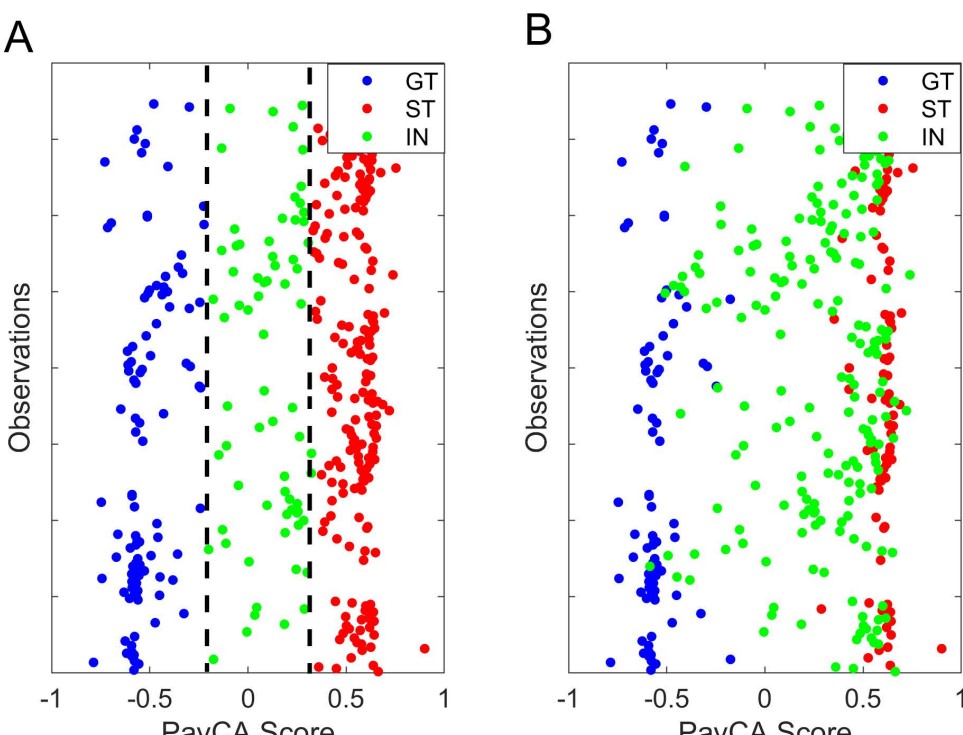

**Fig 6. Classification from 1D vs. 3D k-Means trained on the pooled scores from Days 5 and 6 of the modeling sample. (A)** Pooled Pavlovian Conditioning Approach (PavCA) Index scores from Days 5 and 6, classified with the 1D k-Means (the algorithm was performed directly on the scores). **(B)** Same pooled PavCA Index scores classified with the 3D k-Means. The 3D method does not produce a linear delimitation, prompting for further research.

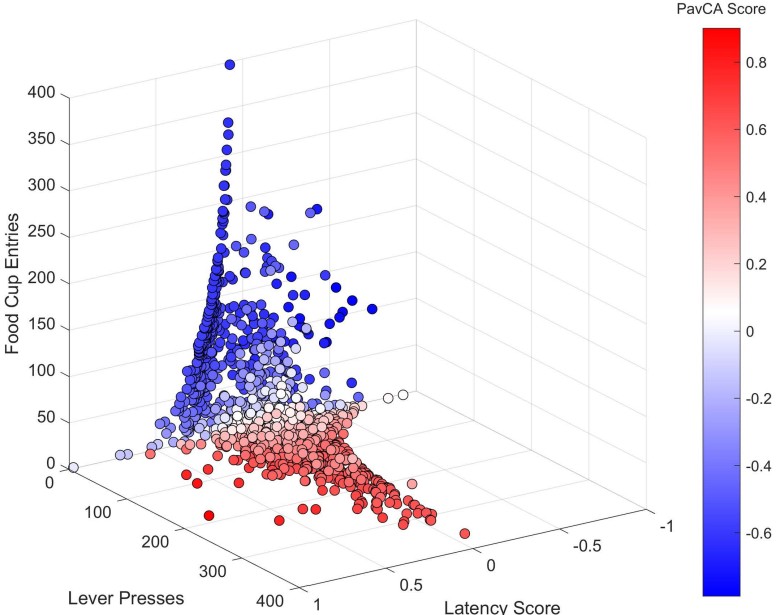

**Fig 7. The relationship between pooled PavCA Index scores from the last 2 days of conditioning and three behavioral parameters.** Pooled data (Days 5 and 6) from the modeling sample of the latency scores, lever presses and food cup entries represented in 3D and color-coded with their associated Pavlovian Conditioning Approach (PavCA) Index scores.

We illustrated the results of each method on both samples in **Fig 8**. It is apparent that using the ±0.5 cutoff values (illustrated by the red line) in a smaller sample (Fig 8B) with a leptokurtic and skewed distribution provides a less suitable classification, as it does not account for the skewed GT group and results in the majority of GT subjects being classified in IN. The $k$-Means method (yellow line) seems to provide excessively inclusive classification in the modeling sample (Fig 8A), whereas the derivative method (green line) seems to provide a middle ground. Yellow asterisks (*) illustrate the centroid value for k-Means, green asterisks illustrate the peak location of the density function (derivative method). The derivative seems the most appropriate classification method on a relatively small sample, while both new methods perform similarly on a larger, pooled sample.

We used McNemar's test for group proportions to evaluate the differences in observations between the three classification methods. As depicted in Fig 4B, the observed frequencies for each method were as follows: k-Means: 18 ST, 8 GT, 8 IN; derivative method = 19 ST, 8 GT, 7 IN; and ±0.5 cutoff: 11 ST, 2 GT, 21 IN. For the ST group, the analysis suggested a significant difference between the ±0.5 cutoff value and the derivative method ($\chi^2 = -0.200$, $p = 0.005$) whereas it suggested no significant difference between the ±0.5 cutoff value and the $k$-Means ($\chi^2 = -0.125$, $p = 0.025$). For the IN group, the results revealed a significant difference between the ±0.5 cutoff value and both $k$-Means classifier and the derivative methods ($\chi^2 = 0.265$, $p < 0.001$ and $\chi^2 = 0.350$, $p < 0.001$, respectively). For the GT group, the analysis suggested no significant difference between the ±0.5 cutoff value and both the $k$-Means and the derivative method (both $\chi^2 = -0.150$, $p = 0.014$).

The classification results provided from both the $k$-Means and the derivative methods were also compared using McNemar's test. The analysis did not show any significant difference in the proportions of observations between the method for the ST and IN groups ($\chi^2 = -0.75$, $p = 0.083$, and $\chi^2 = 0.75$, $p = 0.083$, respectively). However, McNemar's test could not provide statistics for the proportions of GT extracted by the derivative method and the $k$-Means due to high rates of success or failure in the contingency table. Thus, the $k$-Means only showed a significant difference from the ±0.5 cutoffs in the IN

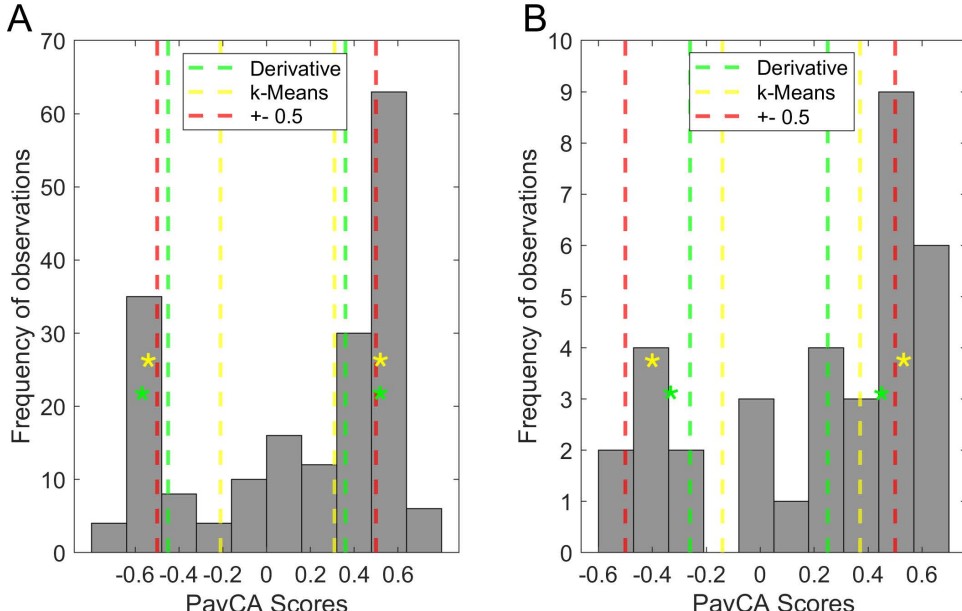

**Fig 8. Mean PavCA Index score distributions and classification values for each method. (A)** Distribution of mean Pavlovian Conditioning Approach (PavCA) Index scores from the last two days of conditioning in the modeling sample ($n = 189$). **(B)** Mean scores in the validation sample ($n = 34$). Colored lines indicate cutoff values determined by each method. Red lines illustrate the ±0.5 cutoff values, yellow lines the k-Means cutoffs, green lines the derivative method. Yellow asterisks (*) illustrate the centroid value for k-Means, green asterisks illustrate the peak location of the density function (derivative method).

group, while the derivative method showed significant differences from the ±0.5 cutoffs in both the ST and IN groups, but not the GT group. Since the test is sensitive to group sizes, and the IN groups are significantly different, any differences in GT group proportions might be considered significant due to their smaller sizes.

## Discussion

Sign-trackers (ST) and goal-trackers (GT) models relate to a subject's tendency to attribute incentive salience to reward cues. To classify subjects into one of the three categories (ST, GT, IN), researchers quantify the PavCA Index score of their subjects and often use ±0.5 cutoff values based on Meyer et al. [13]. Given the common use of arbitrary and variable cutoff values [2,14–18], this study aimed to assess methods that objectively determine appropriate cutoff values for each distribution of PavCA Index scores. The variability of cutoff values indicates heterogeneity in distribution shapes across studies, highlighting the importance of accurate classification based on distribution shape. Our results demonstrate that using the k-Means and derivative methods on the mean scores of the final days of conditioning is effective to classify STs and GTs. These methods enable the classification of subjects into three categories in a standardized manner, yielding data-driven classification even with small samples. We suggest using both methods to compare where the cutoffs fall on the score distribution and select the approach that best reflects the data's structure if they yield different values.

### Population vs. sample distributions and cutoffs

Our analyses operate under the assumption that, in the general population, genetic and neural systems are normally distributed. However, when subjects are exposed to a highly repetitive and unnatural Pavlovian conditioning protocol—designed to reinforce associations between stimuli and responses—their behavioral variability becomes exaggerated,

shifting from a normal distribution to a bimodal one. Importantly, in specific samples of the population, this shift may not occur symmetrically as factors such as environmental stress, early-life experiences, and genetic background can introduce a skew in the initial distribution, which is then maintained or even amplified through conditioning.

In our study, the derivative method yielded cutoff values of -0.45 and 0.36 in the modeling sample (pooled, N = 189), while in the validation sample (N = 34), the cutoffs shifted to -0.26 and 0.25. This reflects the impact of sample composition—pooling subjects from multiple cohorts in our lab resulted in a broader, albeit slightly skewed, bimodal distribution. Conversely, smaller sample sizes tend to produce narrower distributions, leading to subtle variations in cutoff values. These findings reinforce the importance of using an adaptive classification method rather than rigid cutoffs, as population-specific factors can influence the distribution of behavioral phenotypes.

Overall, when applying different classification methods to datasets of varying sizes, we observed notable differences in group proportions. In our large modeling sample, the derivative and k-Means methods yielded similar distributions across groups, with the biggest difference occurring in the size of the IN group. In contrast, the ± 0.5 cutoffs classified the IN group as the largest proportion, shifting the balance of the distribution. These frequency differences became even more pronounced in our smaller validation sample. With the ± 0.5 method, only two subjects were classified as GT, whereas both the k-Means and derivative methods identified eight GTs—demonstrating a substantial discrepancy. Additionally, the k-Means and derivative methods produced nearly identical proportions, differing in the classification of only a single subject. Even if these frequency differences were not statistically significant, they have practical consequences. The ± 0.5 cutoffs, by artificially reducing the GT category, may require training a larger number of subjects to obtain a sufficient sample of GTs for further analyses, thereby raising important ethical implications.

### The application of *k*-Means with *k* = 3 with two clusters

To delve deeper into the results of the k-Means method with k = 3, despite apparent clustering into two groups, we trained the k-Means on the three parameters used to calculate PavCA Index scores. These parameters included the number of lever presses and food cup entries during the training session, along with a latency score reflecting the time taken to press the lever across trials. We examined the relationship between observations across these three parameters and their associated PavCA Index scores. As depicted in Fig 7, there appears to be no clear distinctions or clusters between the lever presses and food cup entry variables. Once again, the interpretation of the k-Means classifier's performance is mitigated by the lack of definite clusters in the data set.

The evolution of the distribution of PavCA Index scores over days of conditioning suggests a tendency for subjects to initially exhibit GT behaviors, with varying degrees of resistance to attributing incentive salience. This is likely due to the pretraining to the food cup, rendering it a familiar stimulus and one with which subjects tend to interact with first. The progression from GT to IN-ST also coincides with a computational model proposed by Lesaint et al. [18], which suggests that some subjects might utilize a model-based strategy, learning the task's structure by employing a systematic approach to accessing the food (GT). Alternatively, subjects could employ a model-free approach, relying on trial and error to evaluate the task's features, which systematically favors lever interactions (ST). Initially, the trial-and-error process in the model-free approach may artificially produce GT-oriented scores. However, as lever presses progressively increase, and food cup entries decrease, the scores shift towards ST.

Evidence from multiple studies [2,13,30] suggests that IN subjects represent a stable psychological trait, as they remain stable after a few days of conditioning. Meyer et al. [19] emphasized that both ST and GT responses are learned behaviors, not innate tendencies, as they are artificially reinforced by repetitive pairing of an unnatural cue and food. While one may argue that IN subjects may transition into ST or GT categories with prolonged exposure to stimuli, research by Robinson and Flagel [30] demonstrates that all three groups remain stable and distinct for up to 22 days of conditioning. This observation challenges the validity of the progressive GT cutoff values extracted by the k-Means classifier. Given the stability of IN subjects, even when using ± 0.5 cutoff values, an appropriate GT cutoff value should not progressively include

more "true" INs over time. In sum, the fact that IN subjects resist transitioning into ST and GT behaviors is an interesting area for further investigation and research.

## Using the classification methods

We recommend that researchers generate their own pooled sample, similar to our modelling sample, by aggregating the last 2 days of PavCA Index scores from previous experiments. This pooled sample should include multiple past datasets, provided that the PavCA protocol was conducted before any experimental manipulation that could influence the distribution. Researchers can then apply both the derivative method and k-Means to determine cutoff values, allowing them to assess the overall classification tendencies within their lab. Since environmental, procedural, or genetic factors may introduce systemic biases, this approach helps establish a lab-specific reference distribution. When analyzing new samples, researchers can compare the extracted cutoff values to those of the pooled sample to contextualize their results within the lab's broader population. Each laboratory functions as its own unique environment with distinct variables, making its subject pool a population in itself. Both the sample's and the population's cutoffs should be reported in studies.

That said, classification methods should be applied individually to each PavCA Index score experiment to avoid reliance on predefined cutoff values, which may not be consistent across cohorts. The lab's population-derived cutoffs should be used only when working with very small sample sizes (i.e., fewer than 10 subjects), where independent distribution analysis is not feasible. Given the variability in distributions due to multiple factors, predefined cutoffs could be adapted to specific periods, colonies, or projects while maintaining a record of the overall lab trend.

As suggested by Meyer et al. [13], the PavCA Index score reflects a complex psychological trait influenced by both genetic predispositions and environmental factors [31]. Given our results and findings from multiple studies indicating stabilization of scores around Days 4 and 5 of training [13,32], we recommend using the derivative method with default bandwidth and an unbounded domain on mean scores that include the last days of conditioning, particularly after relative stabilization. This approach may be more appropriate than using single-day scores, especially for relatively small sample sizes. To ensure methodological rigor, we recommend comparing classification results from both the derivative and k-Means methods before finalizing cutoff values. In our modeling sample, the derivative method produced a larger IN group than k-Means. In such cases, researchers should examine the distribution, compare where the cutoffs from both methods fall, and select the approach that best reflects the data's structure.

We also suggest reporting key distribution metrics, including centroid/peak locations, dispersion values of groups, and the default bandwidth—all of which can be obtained using the code provided in the supplementary materials. Finally, for researchers interested in identifying the most extreme Sign- and Goal-Trackers, data bounds can be adjusted with restrictions (e.g., [-1,1]) to better capture outer boundaries.

## Limits

In an effort to ensure representativeness of a pooled sample such as the foundational study from Meyer et al., our laboratory included both male and female subjects. However, to streamline statistical analysis and avoid unnecessary complexity, we did not calculate distinctions based on sex in the modeling sample. The sexes were separated only in the validation sample to assess the effect of food restriction, as the growth curve had the potential to significantly impact the distribution of scores. Moreover, the results of our validation sample suggest that there are no significant differences between the PavCA Index scores of male and female rats.

In our validation sample, the size of GT groups classified with the ±0.5 cutoff values were small compared to the IN and ST groups. Meyer et al. [13] suggest that ±0.5 cutoffs allow for an even number of classified subjects in each group, but our pooled dataset suggested that more appropriate cutoffs are around ±0.3 to achieve an even ratio, consistent with previous findings [2]. Notably, research papers on ST and GT [2,13,15,33] often use male Sprague-Dawley subjects, whereas we included both female and male Long-Evans subjects. The skewness and kurtosis observed in the distribution of PavCA

Index scores in our laboratory [26] may be attributed to a variety of factors. However, the variables most likely contributing to these deviations are the strain (as mentioned earlier, the Sprague-Dawley strain is more commonly used in ST and GT research) and the vendors [12]. Furthermore, our laboratory location requires transportation of pregnant females by plane, which has been shown to induce stress [34]. Exposure to prenatal stress could therefore have influenced the pups' development [35] and their propensity to attribute incentive salience.

While coinciding with our objectives, it is worth nothing that the $k$-Means method requires predetermined $k = 3$ clusters to find the group edge, which may not accurately reflect the distribution's clusters. As previously mentioned, applying a $k$-Means classifier to extract three groups from a distribution with two apparent clusters can be challenging to interpret and may be technically incorrect. While $k$-Means is popular due to its computational and conceptual simplicity, it has several limitations. For instance, outliers can significantly impact cluster centroids, and determining the optimal $k$-value (number of clusters) can be difficult to determine, particularly in high-dimension data [36]. In our study, the $k$-Means tended to include observations that were part of the IN group, as it could not adequately account for the stabilization observed in the IN group across days. Furthermore, researchers interested in using the $k$-Means method to explore the tendency to attribute incentive salience should be aware that groups need to be sufficiently separated ($\Delta = 4$) and with a minimum of 20–30 observations in each group to ensure adequate power for detecting subgroups [37]. However, the $k$-Means may be a viable option when the distribution is not easily deciphered with the derivative method.

## Conclusion and future research

The ST and GT model enables researchers to quantify a subject's tendency to attribute incentive salience to a cue preceding an unconditional stimulus. Sign-tracking has been considered to be a maladaptive behavior that can be observed across species, including humans [38]. Traditionally, subjects have been classified into ST and GT groups using arbitrarily chosen cutoff values, leading to high variability across literature. Our study demonstrates that with a density estimate and derivative function, appropriate cutoff values can be extracted and tailored to individual distributions. These classification methods cannot eliminate variability in cutoff values, as they are influenced by the underlying distributions. However, they provide a mathematical framework for selecting these cutoffs systematically. While cutoff values may still vary across laboratories and experiments, standardizing the selection process enables meaningful comparisons across studies. This could, in turn, facilitate large-scale, multi-laboratory insights that would be difficult to achieve with purely subjective cutoffs.

We hope that the research community interested in ST and GT will adopt these classification tools, not only for accurate subject classification, but also for deeper exploration in the tendency to attribute incentive salience. An interesting avenue of research could involve investigating how different factors influence classification performance and distribution shapes, or further exploration of IN behaviors. Additionally, the 3D k-Means results highlight the potential for classification using individual variables instead of a compound score. Other nonlinear classifiers, such as Support Vector Machines (SVM) could yield interesting results. Finally, the $k$-Means and the derivative method can also be applied to other types of behavioral data. We hope that researchers encountering similar classification or objectivity challenges will adopt these tools to analyze distributions in their laboratories.

## Supporting information

**S1 Code. MATLAB code for classification analyses.**
(DOCX)

**S2 Fig. Supp. figures.**
(DOCX)

## Acknowledgments

We would like to acknowledge the volunteers and research assistants who participated in the data collection, as well as the help of our colleagues Prof. Alexandre Melanson and Prof. Bradley Harding in the development of the derivative method.

## Author contributions

**Conceptualization:** Camille Godin.

**Formal analysis:** Camille Godin.

**Funding acquisition:** Frédéric Huppé-Gourgues.

**Investigation:** Camille Godin, Frédéric Huppé-Gourgues.

**Methodology:** Camille Godin, Frédéric Huppé-Gourgues.

**Project administration:** Frédéric Huppé-Gourgues.

**Resources:** Frédéric Huppé-Gourgues.

**Software:** Camille Godin.

**Supervision:** Frédéric Huppé-Gourgues.

**Validation:** Camille Godin, Frédéric Huppé-Gourgues.

**Visualization:** Camille Godin.

**Writing – original draft:** Camille Godin.

**Writing – review & editing:** Frédéric Huppé-Gourgues.

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
