## [Decision Letter · Decision Letter 0]

18 Feb 2025

PONE-D-25-02516Behavior Classification: Introducing Machine Learning Approaches for Classification of Sign-Tracking, Goal-Tracking and Beyond

PLOS ONE

Dear Dr. Huppé-Gourgues,

Thank you for submitting your manuscript to PLOS ONE. After careful consideration, the reviewers and I feel that it has merit but will require some revisions to meet PLOS ONE’s publication criteria. We invite you to submit a revised version of the manuscript that addresses the points raised during the review process.

**ACADEMIC EDITOR: **

This is an interesting and important paper that provides an alternative to current analytical approaches to classify sign-tracking and goal tracking phenotypes.Please address carefully the comments from both expert reviewers.Please ensure the sample size for male and female subjects is included in the manuscript, and that sex is included as a variable in the analyses whenever possible.Consider abbreviating the discussion of (or moving to Supplemental materials) data that are not supporting the main argument (i.e., early lever pressing performance and food deprivation state as classifiers).Please include a more specific data sharing plan.

We look forward to receiving your revised manuscript.

Kind regards,

Rita Fuchs

Academic Editor

PLOS ONE

Journal Requirements:

Please ensure that your manuscript meets PLOS ONE's style requirements, including those for file naming. The PLOS ONE style templates can be found at https://journals.plos.org/plosone/s/file?id=wjVg/PLOSOne_formatting_sample_main_body.pdf and https://journals.plos.org/plosone/s/file?id=ba62/PLOSOne_formatting_sample_title_authors_affiliations.pdf

3. Thank you for stating the following financial disclosure: [This research was supported by the Natural Sciences and Engineering Research Council of Canada Graduate Scholarship to C.G. and by the Discovery Grant from the Natural Sciences and Engineering Research Council of Canada (NSERC RGPIN-2018-06285) to F.H-G.].

Reviewers' comments:

Reviewer's Responses to Questions

**Comments to the Author**

1. Is the manuscript technically sound, and do the data support the conclusions?

Reviewer #1: Partly

Reviewer #2: Yes

2. Has the statistical analysis been performed appropriately and rigorously? 

Reviewer #1: Yes

Reviewer #2: Yes

3. Have the authors made all data underlying the findings in their manuscript fully available?

Reviewer #1: Yes

Reviewer #2: Yes

4. Is the manuscript presented in an intelligible fashion and written in standard English?

Reviewer #1: Yes

Reviewer #2: Yes

5. Review Comments to the Author

Reviewer #1: The authors put forward a comparison of k-Means clustering and a derivative-based method for classifying subgroups of Pavlovian conditioning: goal tracking, sign tracking, and indeterminate. The authors do well at outlining the current limitations of how PC subgroups are classified. However, without a ground truth it is difficult to assess the success of any classification method. Therefore the authors rely on arguments of features like stability and a call for future work to carefully consider classification methods rather than following a ‘standard’ or ‘predefined’ cutoff. Although the authors rigorously explore two new methods for classification, there are some details left unclear and theoretical considerations that should be resolved before publication.

Data availability: The authors state that all data will be made available after acceptance, but there is currently no URL, accession number, or DOI associated with the data.

Major points:

Lines 121-123: Authors argue that the Meyer cutoff values result in uneven numbers of subjects across groups and thus will complicate hypothesis testing. It is unclear that the statistical inconvenience warrants altering the metric which may be accurately classifying subjects (i.e., is there a principled reason to believe that the subgroups should have the same number of subjects?)

Line 288: The authors use the ksdensity function to approximate their density probabilities. It seems that the assumption is that the data will be bimodal and not normally distributed. In this case the bandwidth chosen for the function could have a large impact on the results. What bandwidth did the authors use and why? If they tried other bandwidths, did that have meaningful shifts in their results? Last, the data is bounded [-1 1] but it is not clear if the authors used a bounded support with ksdensity. The code provided suggests that the default bandwidth and support were used.

Lines 343-347: I’m unclear what is being said here. Do the authors mean that Days 1 and 2 combined were different from all other days individually, but were themselves not different from each other? Likewise, were days 3 and 4 individually different from day 6? Perhaps a visualization of these data such that significance bars can be seen would help clarify the differences (e.g., violin plots with lines and asterisks above the data significantly different from each other). Last, it is not clear how these individual differences were detected; post-hoc t-tests?

Throughout the analysis the authors sometimes pool data (e.g., line 354) and sometimes take the average (e.g., line 367). It would be helpful to provide a rationale for when one method is more appropriate than the other. Why was pooled data used for the k-means clustering, but not the derivative method?

The frequencies of observations (lines 412-431) are reported in terms of whether significant differences were found, but the actual frequencies are not reported in either a table or figure.

Lines 449-450: the authors claim that the two methods “...primarily affect the classification of individuals located in close proximity to group boundaries”. What is this based on?

Figure 5: my interpretation of this figure is that the authors are showing the mean and 95% CI PCA score for each group after classification with the three methods. If this is the case, then I am not clear on how there could be PCA scores classified as, for example, GT by the K-means clustering on day 1 if the K-means cutoff value for GT on day 1 is <-0.44 (table 1) while the 95% CI goes above -0.4. If these classifications were obtained from the pooled clustering, then the problem seems even worse as that cutoff was <-0.22 which encompasses likely all the IN data and a large chunk of ST.

The authors try a 3D k-means clustering, but state “...there were not three definite clusters…” (lines 480-481). How was this determined? If only visually, could one not make a similar argument for the distributions seen in Figures 5 and 6? Purely visually I might say there are perhaps 2 groups instead of 3. Further, did the authors try any non-linear methods like SVM to classify either the computed PCA scores or the 3 variables in the 3D clustering?

The authors claim that the derivative method “...seems to provide a middle ground.” (line 510). It would be nice to see some exploration of the theoretical basis of this. Given two normal distributions, what percent of the data will fall in a tail defined by the derivative method? In other words, assuming normalcy, the first derivative will always be at the same z-score (±1) and thus cut off about 16% of the distribution. Is there a principled reason for believing that this cutoff will better approximate biologically relevant subgroups? Granted, these assumptions won’t hold in non-normal distributions which seem to be prevalent in these data. What do the authors think will be the consequence in bimodal distributions? Will the derivative find approximate half-way points between the two underlying distributions? Is that what we should expect to best separate the groups?

A consistent motivation the authors give is the “high variability” (e.g., line 705) in the literature in defining subgroups in Pavlovian conditioning. However, it does not seem that their methods will necessarily solve this problem. Especially as the authors acknowledge that their is likely no one best method, even within a lab, and that cutoff values should be determined for each cohort. They do mention that PCA scores tend to stabilize as sample sizes increase and on later conditioning days, but there should be some acknowledgement that either the k-means clustering or derivative method may still lead to variability in cutoff values across labs and experiments.

Minor points:

Line 65: By “positive outcome” are the authors referring to valence, an outcome in which something is added, or something else?

Line 145: “...PCA scores tend to distribute in a bimodal, U-shaped fashion…” are there citations to back this up? The authors themselves later argue that the bimodality may depend on day of conditioning and number of subjects.

Line 175: The number of each sex used for the modeling sample is not stated.

Line 175: What other experiments did these rats contribute to? If they are published, provide citations, if not, provide some details to allay concerns that the other experiments may influence Pavlovian conditioning.

Missing details on validation sample: were these rats also used in other experiments? were they trained during the same time frame as the modeling sample (June 2021 - July 2022)?

Line 190: what is the breakdown by sex of the two cohorts in the validation sample?

Line 203: assuming that each operant box only had one lever being used, were rats randomly assigned to a left or right lever in equal numbers?

Line 234: “Each contact…” do the authors mean any kind of contact or a full lever press?

Line 235-237: “We noted…the total number of consumed pellets” How as this determined? As a subtraction of pellets delivered and pellets remaining in the dispenser?

Line 243: What do the authors mean by “response” if not lever presses or food cup entries?

Line 265: Earlier in the methods the validation sample size was given as 34 rats (line 177)

Line 341: Why was sex not considered a factor in the analysis of the modeling sample?

Table 1: a row should be added for the results of the pooled days 1–6 k-Means clustering.

Line 433: I’m surprised the authors did not note that despite both methods extracting “unstable cutoff values” that there were also very similar values.

Lines 466-467: Says “...pooled data from all six days…” while the figure legend for figure 6 specifies “...Days 5 and 6…” (line 475).

Line 471: “...delimited around ±0.3…”; doesn’t Table 1 say >0.31 and <-0.21 for pooled days 5 and 6?

Line 497: “The validation sample did not include two parabolas…”; please provide a figure showing this.

Lines 713-714: “These variations may be related to subclusters of PCA Index scores and could be further explored with cluster analysis.” Are the authors suggesting more than 3 clusters may exist? If so, how do they propose they are defined/explored given that k-means was already having issues with reliably finding 3 clusters?

Reviewer #2: The manuscript by Godin and Huppe-Gourgues describes two approaches for behavioral characterization of individual variability. They apply these approaches to the sign-tracker/goal-tracker model and compare them to the classification method that has traditionally been applied to this model, the Pavlovian conditioned approach (PCA) index. The results suggest that the newly described methods of classification are effective tools for identifying sign-trackers and goal-trackers and may be especially useful for small sample sizes. Although the differences between these approaches and the traditional PCA index method were not striking, the newly described approaches could potentially provide a more standardized classification framework. This is a well-written, timely, and important manuscript that could be improved with attention to the points raised below.

Line 28: The authors define ST and GT as sign-tracking and goal-tracking, respectively, but should consider defining as “sign-trackers” and “goal-trackers” to keep consistent with the relevant literature.

Line 32: The authors might consider changing “PCA Index” to “PavCA Index” so the readers do not confuse this terminology with principal components analysis (PCA).

Line 66: The authors should note that predictive cues can act as incentive stimuli, but they do not always and often just for a subset of individuals (i.e., sign-trackers).

Line 95-96: It is a misnomer to describe the PCA Index as a ratio of head entries to lever presses, when it is actually a composite index score, as described in the subsequent sentence.

Line 166: It doesn’t seem all that common to food-restrict animals prior to assessing Pavlovian conditioned approach behavior in the sign-tracker/goal-tracker literature. Further, the effects of food-restriction on sign-tracking/goal-tracking behavior have been reported (e.g., Anderson et al., 2013 PMCID: PMC3845669 DOI: 10.1016/j.bbr.2013.09.021; Boakes Chapter, 1977). Given this, and the fact that the food-restriction data does not add much to the current manuscript, it is recommended that this dataset be removed from the current manuscript. Alternatively, it should be presented at the end of the manuscript, not at the beginning (see more below).

Lines 173-175: The number of males and females in the n=189 sample should be specified.

Line 201: What does PLA stand for?

Line 264: It is stated here that the validation data n=58; whereas earlier in the methods (line 177) is states that n-34 for this sample. Please clarify.

Figure 1: As suggested above, it is suggested that the effects of food availability either be removed or moved to a different section of the manuscript. It would be beneficial to the reader and the models if the initial data presentations focused on the typical Pavlovian conditioned approach metrics illustrated and analyzed with sex and session as variables. Further the data should be illustrated to better align with the descriptions in the text. For example, if sex differences are being analyzed/reported, then males and females should be shown in the same graphs.

Lines 365-371: It is not clear why the new approaches were applied to early training days, given that the behavioral phenotypes don’t emerge or become stable until later training days. Further explanation for this approach is needed. In addition, further description of what the centroids and point-to-centroid distances refers to would be helpful. Finally, if there were a way to graphically illustrate the k-means cluster data (i.e., that shown in Table 1), that might also be helpful to the reader (as was done for the derivative method).

Lines 413-423: It would be helpful to illustrate the data described here as histograms and to state the actual number of observations that differed between the different classification methods.

Lines 430-436: Further description of the “center” value and how that should be interpreted is warranted. It would be helpful if a few sentences were added to this section to better explain what these findings actually mean.

Lines 483-485: What does latency score refer to here? Is it the latency difference score between the lever and food cup?

Lines 508: It is not clear on what basis the authors claim that the +/- 0.5 cutoff value provides “less optimal classification”. What makes this approach less optimal in this case?

Figure 8: Do the data shown in Figure 8B include both food groups? It would be ideal if the validation group was tested under the same conditions as the model group.

Lines 522-529: As described above, it would be helpful to state how many observations (i.e., animals) were differentially classified based on the different methods.

Lines 553: What does “accurate classification” mean in this case?

Line 564: Which trends supported the sex differences referred to here?

Lines 565-570: Other studies have suggested that female rats tend to be skewed more towards sign-trackers than males rats (e.g., Hughson et al., 2019, PMCID: PMC6382850 DOI: 10.1038/s41598-019-39519-1). Thus, it seems like it would be important to assess the models separately in each sex and see how each compares.

Line 618: It is likely the pretraining to the food cup that skews early behavior towards goal-tracking, and this should be noted.

Lines 661-662: Further guidance is needed to clarify what the authors suggest should be done to select cutoff methods using one approach versus another.

Line 713: It is not clear what the authors are referring to in the Lovic et al. paper (reference 39) wherein it seems that STs were generally more impulsive on tests of impulsive action relative to GTs.

6. PLOS authors have the option to publish the peer review history of their article (what does this mean? ). If published, this will include your full peer review and any attached files.

**Do you want your identity to be public for this peer review?** For information about this choice, including consent withdrawal, please see our Privacy Policy .

Reviewer #1: No

Reviewer #2: No

---

## [Author Response · Author response to Decision Letter 1]

4 Apr 2025

April 3rd, 2025

Dear Dr Fuchs,

We sincerely appreciate the reviewers’ thoughtful and constructive feedback, which has helped us refine our manuscript significantly. In response to their comments, we have made several key revisions and adjusted the manuscript’s main focus accordingly.

First, we removed the daily classification of the first few days of conditioning, as we recognize that these early classifications were unreliable and not central to our conclusions. While these analyses were initially conducted to understand how the K-means and derivative methods performed classification on the final days of conditioning, we have decided that they do not warrant primary emphasis.

Additionally, we have removed the analysis of food effects from the main text, as it does not represent a core result of the study. However, we can provide these results in the supplementary material for transparency. Similarly, we have eliminated the pooled classification results for K-means, as they were part of our early exploratory analyses but do not substantively contribute to our conclusions. After careful consideration, we determined that pooling daily data for classification is not an approach we would recommend.

Furthermore, we have clarified that our modeling sample was constructed to reflect the approach used in the foundational study by Meyer et al., which combined samples from multiple cohorts and colonies. Just as in their study—published in the same journal—we do not have immediate access to sex information for our modeling sample, as it is a combined dataset derived from numerous experiments conducted over the years by different students in our lab. However, we have made sure to explicitly represent sex differences (or the lack thereof) in the first figure, ensuring transparency in our findings. Our findings support the hypothesis that larger pooled samples tend to approximate population-level distributions, whereas smaller subsamples do not exhibit the same degree of expansion and symmetry. Note that in this context, pooling refers to t combined cohorts’ trends based on the averaged last days of conditioning, rather than data from all their conditioning days.

We have established a data-sharing plan. The dataset, along with the MATLAB code used for analysis, will be made available on our Open Science Framework (OSF) platform, and we will provide the DOI for free access. The uploaded materials will include:

• An Excel sheet containing the PavCA Index scores for both the modeling and validation samples, organized by conditioning days.

• For the validation sample, additional columns specifying sex and food group.

• A README document providing details on the dataset structure and instructions on how to use the code for replication and further analysis.

We are very grateful to the reviewers for their insightful comments, which have allowed us to refine our interpretation and improve the manuscript extensively. We look forward to your consideration and hope that the revised version meets the journal’s expectations.

You will find below our responses to reviewers’ comments.

Sincerely,

Frédéric Huppé-Gourgues

Reviewer #1

Major points:

Lines 121-123: Authors argue that the Meyer cutoff values result in uneven numbers of subjects across groups and thus will complicate hypothesis testing. It is unclear that the statistical inconvenience warrants altering the metric which may be accurately classifying subjects (i.e., is there a principled reason to believe that the subgroups should have the same number of subjects?)

Great question. Our concerns with the Meyer cutoff values go beyond statistical convenience; they have practical implications for both subject recruitment and classification validity, which we have now clarified in the manuscript.

First, applying the Meyer cutoffs can lead to highly unbalanced group sizes depending on the distribution (e.g., 2 GTs, 11 INs, and 21 STs in our validation sample). This imbalance is not just a minor statistical issue—it significantly impacts the feasibility of research. Since researchers typically require a minimum number of subjects per subgroup, an extreme imbalance means that substantially more subjects must be screened and trained to obtain an adequate sample of GTs, increasing resource demands and raising important ethical implications. This has been added to the discussion (lines 566-580).

Second, there is a principled reason to expect that the subgroups should be of similar size. Meyer et al. originally described GT and ST phenotypes as emerging from genetic and environmental factors, which, if normally distributed in the population, should produce roughly equal numbers of GTs and STs. However, the PCA method used to define these subgroups artificially enforces a bimodal distribution, exaggerating extreme scores from the original normal distribution. If the underlying distribution is slightly skewed due to genetic and environmental influences, this further distorts the natural proportions. As a result, Meyer’s cutoffs tend to overrepresent STs while underrepresenting GTs—not because GT-like individuals do not exist, but because the cutoff criteria fail to capture them.

Our approach accounts for this skewness, ensuring that classification better reflects the expected distribution of phenotypes rather than being an artifact of the analytical method. While natural variation in subgroup sizes is expected, extreme imbalances suggest that the cutoffs may not optimally capture the underlying biology.

Fitzpatrick et al. (2013) analyzed PCA Index Score distributions from 557 subjects across different vendors (cited on line 90), showing that some samples were skewed toward ST while others were skewed toward GT. Similarly, Meyer et al.'s foundational study—based on 1,878 rodents from multiple vendors—showed an almost perfectly bimodal distribution (Figure 6), suggesting that in larger populations, scores tend to distribute bimodally with relatively even group sizes. However, in smaller samples, these distributions naturally exhibit skewness, leading to uneven subgroup sizes. The standard 0.5 cutoff works well for large, population-wide samples but does not always apply to smaller datasets where bimodality is present but imperfectly balanced.

In summary, these population insights suggest that GTs and STs distribute bimodally with roughly even proportions in large samples. However, in more realistic, smaller datasets, natural skewness emerges, requiring cutoffs that adapt accordingly. Our method offers a principled, mathematical approach to this adaptation, reducing subjective bias while maintaining consistency with the underlying biological distributions. We have expanded on this in the manuscript from line 118-131.

Line 288: The authors use the ksdensity function to approximate their density probabilities. It seems that the assumption is that the data will be bimodal and not normally distributed. In this case the bandwidth chosen for the function could have a large impact on the results. What bandwidth did the authors use and why? If they tried other bandwidths, did that have meaningful shifts in their results? Last, the data is bounded [-1 1] but it is not clear if the authors used a bounded support with ksdensity. The code provided suggests that the default bandwidth and support were used.

We used the default bandwidth, which is based on Silverman’s rule of thumb, as it provides a data-driven estimate without requiring prior assumptions about the distribution. To assess robustness, we tested different bandwidth values and found that while the smoothness of the density estimates varied, the overall distribution shape, including bimodality (if present), remained consistent. We added details in the manuscript regarding this (lines 303-304), and we modified the code, so it provides the generated default bandwidth. For our modeling data set, the default bandwidth was 0.14, and 0.18 for our validation dataset.

Regarding the data bounds, you are correct that our data is bounded within the range [-1,1]. By default, ksdensity estimates density over an unbounded domain, which can result in minor density spillover beyond this natural range. When restricting ksdensity to the [-1,1] interval, the density function becomes condensed and slightly distorted, leading to small shifts in the estimated cutoff values. However, since our focus is on the inner boundaries of the GT and ST groups rather than their outer limits, this spillover does not impact our key findings. The classification remains robust, and the biological interpretation of subgroup distinctions is unaffected.

Lines 343-347: I’m unclear what is being said here. Do the authors mean that Days 1 and 2 combined were different from all other days individually, but were themselves not different from each other? Likewise, were days 3 and 4 individually different from day 6? Perhaps a visualization of these data such that significance bars can be seen would help clarify the differences (e.g., violin plots with lines and asterisks above the data significantly different from each other). Last, it is not clear how these individual differences were detected; post-hoc t-tests?

We have clarified lines 341-345 to make the comparisons more explicit. Additionally, we confirm that individual differences were detected using Tukey post-hoc tests.

Throughout the analysis the authors sometimes pool data (e.g., line 354) and sometimes take the average (e.g., line 367). It would be helpful to provide a rationale for when one method is more appropriate than the other. Why was pooled data used for the k-means clustering, but not the derivative method?

Fair point, we have decided to remove the pooled data from the manuscript as it does not contribute to the overall results substantially. Initially, we explored both approaches to determine which would be most appropriate for different analyses.

For the derivative method, pooling data was not suitable because the early days of conditioning do not exhibit a bimodal distribution. When pooling across days, the resulting density curve is even further from a bimodal distribution than the skewed distribution obtained from the last two-day average. Averaging aligns with common practices in the literature, allowing for better comparability with previous studies. Additionally, with the derivative method, a smaller sample size can lead to a less smooth density function, though this can be adjusted via bandwidth selection, which we have now clarified.

The frequencies of observations (lines 412-431) are reported in terms of whether significant differences were found, but the actual frequencies are not reported in either a table or figure.

Added the frequencies (lines 410-413 for modeling sample, 514-516 for the validation sample).

Lines 449-450: the authors claim that the two methods “...primarily affect the classification of individuals located in close proximity to group boundaries”. What is this based on?

Added some details (lines 438-440).

Figure 5: my interpretation of this figure is that the authors are showing the mean and 95% CI PCA score for each group after classification with the three methods. If this is the case, then I am not clear on how there could be PCA scores classified as, for example, GT by the K-means clustering on day 1 if the K-means cutoff value for GT on day 1 is <-0.44 (table 1) while the 95% CI goes above -0.4. If these classifications were obtained from the pooled clustering, then the problem seems even worse as that cutoff was <-0.22 which encompasses likely all the IN data and a large chunk of ST.

You are correct in noting the potential inconsistency in classification based on the provided cutoff values. The reasoning behind applying the classifiers to individual days rather than the mean of the last two days was to gain insights into how groups progressed toward their final classification and to better understand the classifiers’ limitations. However, after further reflection, we have revised the manuscript to focus on the most relevant results – classification based on the last two days of conditioning – rather than pooled or individual-day analyses.

The authors try a 3D k-means clustering, but state “...there were not three definite clusters…” (lines 480-481). How was this determined? If only visually, could one not make a similar argument for the distributions seen in Figures 5 and 6? Purely visually I might say there are perhaps 2 groups instead of 3. Further, did the authors try any non-linear methods like SVM to classify either the computed PCA scores or the 3 variables in the 3D clustering?

This was determined purely visually, and we agree that one could potentially argue for two groups rather than three. This aligns with the idea that a normally distributed set of behaviors may be exacerbated into a bimodal distribution, which could suggest two distinct groups (two distributions: GT and ST). However, researchers often aim to retain three groups, as INs appear to represent a unique group that is distinct from both GT and ST. While we have not explored non-linear methods like SVM, we appreciate the suggestion and have added it to the conclusion as a potential avenue for future work (lines 712-714). The primary goal of this analysis was to provide an initial attempt at classifying subjects using the individual variables rather than the compound scores, and it seems there is more to investigate in this area.

The authors claim that the derivative method “...seems to provide a middle ground.” (line 510). It would be nice to see some exploration of the theoretical basis of this. Given two normal distributions, what percent of the data will fall in a tail defined by the derivative method? In other words, assuming normalcy, the first derivative will always be at the same z-score (±1) and thus cut off about 16% of the distribution. Is there a principled reason for believing that this cutoff will better approximate biologically relevant subgroups? Granted, these assumptions won’t hold in non-normal distributions which seem to be prevalent in these data. What do the authors think will be the consequence in bimodal distributions? Will the derivative find approximate half-way points between the two underlying distributions? Is that what we should expect to best separate the groups?

In our analysis, we are working under the assumption that the population exhibits a normal distribution of genetic and neural systems. When forced into a highly repetitive and unnatural Pavlovian conditioning protocol, which forces an association between stimuli and responses, we observe a shift from this normal distribution to a bimodal one. This shift occurs because the experimental design tends to enhance the behaviors of subjects at the extremes of the normal distribution. Essentially, the experimental protocol manipulates the natural variability in a way that accentuates the extremes, resulting in a bimodal distribution of responses. In samples of the population, the normal distribution that partially stems from the state of the nervous system may be skewed towards one end (i.e. sign-tracking due to stress). This skewness is then maintained or amplified under the Pavlovian conditioning protocol, and as a result, the distribution remains roughly bimodal but with a skewed pattern.

When we run the derivative method on a random bimodal distribution (peaks at -0.6 and 0.6 with [-1,1] boundaries, and SD of 0.2), ~38.3% of the values fall before the first cutoff (GTs), and ~40.7% fall after the second cutoff (STs), leaving ~20.7% as IN. In our validation sample, 25.53% fell in GTs, and 52.94% in ST, leaving 23.53% in IN. When using the +-0.5 Meyer cutoff on our validation sample, only 5.89% of observations fell in the GTs (2/34).

Thus, the derivative method captures the points where the bimodal distribution “naturally” separates into distinct subgroups. The Meyer cutoffs (-0.5 and 0.5) qualitatively correspond to the minima and maxima points of the density curve to identify GTs and STs. Although clustering subjects in 3 groups might not be definitive (there could be more than 3 groups), we extend this approach because it allows to partition the population into biologically relevant groups that have been consistently identified in t

---

## [Editor Report · Decision Letter 1]

16 Apr 2025

Behavior Classification: Introducing Machine Learning Approaches for Classification of Sign-Tracking, Goal-Tracking and Beyond

PONE-D-25-02516R1

Dear Dr. Huppé-Gourgues,

We’re pleased to inform you that your manuscript has been judged scientifically suitable for publication and will be formally accepted for publication once it meets all outstanding technical requirements.

Kind regards,

Rita Fuchs

Academic Editor

PLOS ONE

Additional Editor Comments (optional):

The authors did a great job addressing the comments of the reviewers. I think this paper contributes an important new approach to studying the sign-tracking/goal-tracking phenomenon.
---

## [Editor Report · Acceptance letter]

PONE-D-25-02516R1

PLOS ONE

Dear Dr. Huppé-Gourgues,

I'm pleased to inform you that your manuscript has been deemed suitable for publication in PLOS ONE. Congratulations! Your manuscript is now being handed over to our production team.

Kind regards,

on behalf of

Dr. Rita Fuchs

Academic Editor

PLOS ONE